# Could microRNA Analysis Help in the Management of Medullary Thyroid Cancer?

**DOI:** 10.3390/cancers17040629

**Published:** 2025-02-13

**Authors:** Beatriz Febrero, Inmaculada Ros-Madrid, Beatriz Revilla-Nuin, Miriam Abellán, Juan José Ruiz-Manzanera, Joaquín Gómez, José M. Rodríguez

**Affiliations:** 1Endocrine Surgery Unit, General Surgery Service, Hospital Clínico Universitario Virgen de la Arrixaca, 30120 Murcia, Spain; 2Digestive, Endocrine and Abdominal Organ Transplantation Surgery Unit, Instituto Murciano de Investigación Biosanitaria Pascual Parrilla (IMIB Pascual Parrilla), 30120 Murcia, Spain; 3Department of Surgery, University of Murcia, 30120 Murcia, Spain; 4Endocrinology Service, Hospital Clínico Universitario Virgen de la Arrixaca, 31120 Murcia, Spain; 5Genomics Platform of Instituto Murciano de Investigación Biosanitaria (IMIB-Pascual Parrilla), 30120 Murcia, Spain; 6Endocrine Surgery Unit, Department of General and Digestive Surgery, Instituto de Investigacióon Hospital Universitario La Paz (IdiPAZ), 28046 Madrid, Spain, 28046 Madrid, Spain

**Keywords:** medullary thyroid carcinoma, microRNAs, progression, aggressiveness

## Abstract

Understanding the pathophysiology of medullary thyroid cancer (MTC) may help to improve diagnosis, staging, and follow-up planning based on the identified prognostic factors. Currently, there is little evidence on the genomic investigation of microRNAs in patients with MTC. This study aims to determine the overexpression of microRNAs that may increase the diagnostic sensitivity of MTC. Further objectives include evaluating their association with clinical characteristics of aggressiveness, thereby guiding therapeutic decisions and follow-up strategies.

## 1. Introduction

Medullary thyroid carcinoma (MTC) is a neuroendocrine tumor that originates from the parafollicular cells of the thyroid gland. Although it is a rare disease, with an estimated prevalence of 1 in 14,300 individuals, MTC is characterized by a more aggressive phenotype compared to DTC especially when distant metastases are present, with a 5-year survival rate of 74% for advanced tumors [1].

Historically, the measurement of calcitonin (Ct) levels has been the primary tool for the diagnosis, follow-up, and assessment of disease progression in MTC patients. During follow-up, the doubling times of calcitonin and CEA are used as additional markers to evaluate disease progression. Furthermore, the TNM staging system enables the classification of MTC and is closely associated with its prognosis [2].

In recent years, the characterization of additional genomic features, such as the transcriptome, methylome, and miRnome, has emerged as a potential avenue for identifying new molecular markers associated with specific clinical features of MTC [3,4,5].

MicroRNAs (miRNAs) are small non-coding RNA molecules that function as potent regulators of gene expression, acting at the post-transcriptional level and modulating genome-encoded messenger RNAs. Increasing evidence supports the importance of miRNAs in thyroid carcinogenesis, with potential roles as diagnostic and prognostic biomarkers for thyroid neoplasms, including both papillary [6,7,8] and medullary carcinomas [2,3,4,6,7,8,9].

However, most studies focus on papillary thyroid carcinoma [6,7,8], while only a few studies have analyzed these markers in MTC [3,4,10,11,12,13,14,15,16,17,18]. Several miRNAs have been studied, including miR-200b, miR-200c, miR-21, miR-183, miR-375, miR-144, miR-224, miR-34a, miR-146b, miR-29, miR-7, miR-193, miR-222, miR-221, and miR-181b, among others. Among these, miR-200c-3p, miR-144-3p, miR-34a-5p, miR-183-5p, and miR-375-5p appear to be associated with greater tumor progression, including lymph node involvement [3,4,13,15,16,17,18]. Additionally, some studies have explored the differences between sporadic and familial MTC [4,16,17,18,19].

The aims of this study are as follows: (1) to evaluate the expression of different miRNAs in patients with MTC compared to corresponding (paired) healthy thyroid tissue, and (2) to determine the expression of these miRNAs in relation to epidemiological and clinical variables that may be associated with progression or the familial nature of MTC.

## 2. Materials and Methods

### 2.1. Study Population

Twenty-five patients diagnosed with MTC via histological confirmation, operated on by the same surgical team and under follow-up in the Endocrine Surgery Unit Clinic of the Hospital Clínico Universitario Virgen de la Arrixaca (HCUVA) during 1998–2016, were included. Patients of both sexes, including both sporadic and familial cases, as well as those with lymph node involvement, persistence, recurrence, and/or metastasis during follow-up, were selected.

### 2.2. miRNA Expression Analysis

This study was conducted at the Genomics Platform of the Instituto Murciano de Investigación Biosanitaria-Pascual Parrilla (IMIB-Parrilla).

The retrotranscription reaction (RT) was performed using miRCURY LNA RT kit (#339340 Qiagen, Hilden, Germany) with 200 ng of total RNA. Gene expression analysis was carried out by real-time PCR using the miRCURY LNA miRNA SYBR Green PCR kit (#339346, Qiagen) in a QuantStudio 5 kit (Applied Biosystem, Waltham, MA, USA).

The expression levels of five miRNAs (miR-34a-5p, miR-144-3p, miR-200c-3p, miR-375-5p, and miR-183-5p) were quantified in the 25 histological thyroid samples. All primer sets were commercially available from Qiagen (miR-34a-5p (#YP00204486), miR-144-3p (#YP00204754), miR-200c-3p (#YP00204482), miR-375-5p (#YP02105850), and miR-183-5p (#YP00206030)). A comparative analysis was performed using the geometric mean of two reference miRNAs: hsa-miR-103a-3p (#YP00204063) and hsa-miR-191-5p (#YP00204306). These reference miRNAs were selected for their greater stability, regardless of tissue type or storage conditions, as well as for their widespread use in similar studies [20]. Samples were analyzed in triplicate, and the reaction conditions for real-time PCR followed the manufacturer’s specifications provided in the kit.

To determine miRNA expression, the Ct or Cq method [21] was applied, following these steps:

First, the difference between the target gene and the reference gene was calculated:ΔCt (sample) = Ct (target gene) − Ct (reference gene)

Next, the difference between the experimental sample and the paired healthy control was computed (ΔΔCt).ΔΔCt (sample) = ΔCt (experimental sample) − ΔCt (paired healthy control)

Finally, relative changes in gene expression were determined using the formula 2^ΔΔCt^. Additionally, relative expression in tumor and paired healthy tissue was assessed using the 2^−ΔCt^ formula [20].

### 2.3. Clinical Variables

Epidemiological variables: sex and age at the time of surgery.

Clinical variables: preoperative Ct levels, sporadic or familial MTC, and type of surgery.

Histological variables: average tumor size and lymph node involvement (number of nodes removed and number of positive nodes).

TNM and disease stage: T (tumour size), N (nodal involvement), and M (distant metastasis).

During follow-up, patients were classified based on their Ct levels at 6–12 months and the presence of cure or persistence. Additionally, until the end of the follow-up in 2023, the following clinical outcomes were evaluated: structural recurrence, distant metastases, and/or death.

Persistence was defined as elevated Ct and CEA levels within the first 6 months postoperatively, while recurrence was defined as their appearance after this period. Biochemical recurrence was identified in the absence of structural disease on imaging techniques.

Disease-free survival (DFS), defined as the time elapsed from surgery to the detection of recurrence, was also evaluated in patients with recurrence.

### 2.4. Data Analysis

Data analysis was performed using IBM SPSS Statistics (version 29). Descriptive statistics included the mean, median, standard deviation, and range for quantitative variables, and percentages for qualitative variables.

The Shapiro–Wilk test was used to assess the normality of miRNA expression data. Median comparisons were conducted using the Mann–Whitney U, Kruskal–Wallis, and Wilcoxon tests. Spearman’s coefficient was employed to assess correlations. A *p*-value of <0.05 was considered statistically significant.

## 3. Results

### 3.1. Clinical Overview of the Series

Twenty-five patients with MTC were included (Table 1). Sixty percent were women (*n* = 15). The mean age at diagnosis was 41.84 years. Fifty-six percent of the patients (*n* = 14) had familial MTC, associated with MEN2a syndrome.

The mean preoperative Ct and CEA level was 1556.57 pg/mL and 30.6 ng/mL, respectively, with a median of 949 pg/mL and 18.25ng/mL, respectively. Regarding the type of surgical intervention, 28% (*n* = 7) underwent total thyroidectomy (TT), 8% (*n* = 2) underwent TT with unilateral central neck dissection (CND), 4% (*n* = 1) underwent TT with bilateral CND, 4% (*n* = 1) underwent TT with unilateral central and ipsilateral jugular dissection, 12% (*n* = 3) underwent TT with bilateral central and ipsilateral jugular dissection, and 44% (*n* = 11) underwent TT with bilateral central and bilateral jugular dissection.

The median tumor size was 23.92 mm [5–70 mm]. Lymphadenectomy was performed in 72% of cases (*n* = 18), with lymph node involvement observed in 60% (*n* = 15) of patients. On average, 29.19 lymph nodes were removed, with a mean of 7.13 positive nodes.

Regarding TNM classification (Table 2),

T classification: 16% (*n* = 4) were classified as T1a, 28% (*n* = 7) as T1b, 36% (*n* = 9) as T2, and 16% (*n* = 4) as T3.

N classification: 12% (*n* = 3) were classified as N1a, 48% (*n* = 12) as N1b, 12% (*n* = 3) had no lymph node involvement (N0), and in 28% (*n* = 7) no lymphadenectomy was performed (Nx).

Regarding tumor stage, 20% (*n* = 5) were stage I, 12% (*n* = 3) were stage II, 8% (*n* = 2) were stage III, and 56% (*n* = 14) were stage IV.

Regarding disease progression, at one year after surgery, 68% (*n* = 17) had persistent disease, 24% (*n* = 6) were cured, and 8% (*n* = 2) had unknown outcomes.

At follow-up, 52% (*n* = 13) had structural disease, including 11 patients with biochemical persistence one year after surgery; outcomes for the remaining patients were unknown. The mean calcitonin and CEA levels at 6–12 months of follow-up were 952.63 pg/mL and 6.98 ng/mL, respectively, with a median 49 pg/mL and 2.7 ng/mL. The mean DFS was 115.80 months. During follow-up, 20% (*n* = 5) of patients died: three patients due to MTC progression, one from urothelial carcinoma progression, and one from ischemic cardiovascular disease.

### 3.2. Results of the Fold-Change Analysis of miRNA Expression

The fold change in miRNA expression was highest in miR-183-5p, followed by miR-375-5p, with values of 7.666 and 2.226, respectively (Figure 1 and Table 3). Tumor tissue showed a significantly higher fold-change expression of miR-183-5p and miR-375-5p and a lower expression of miR-144-3p compared to paired healthy tissue (*p* < 0.05) (Table 4).

Expression differences were evaluated according to clinical variables (Table 5).

As for miR-200c, no significant differences were observed with respect to age at the time of surgery, sex, heritability, preoperative Ct levels, or tumor size (*p* > 0.05). A trend towards lower fold-change expression was noted in patients with lymph node involvement (0.97 vs. 1.55 vs. 1.26; *p* = 0.204) and in those with stage III/IV compared to stage I/II (0.96 vs. 1.36; *p* = 0.159). Similarly, patients with structural disease showed lower miR-200c-3p fold-change expression compared to others (0.97 vs. 1.25; *p* = 0.115). A moderate positive correlation was found between DFS and miR-200c-3p expression (Spearman’s *p* = 0.40, moderate correlation; *p* = 0.048) (Figure 2). No other differences were found in the rest of the clinical variables.

With respect to miR-144-3p, miR-34-5p, miR-183-5p, and miR-375-5p, no significant differences were found with respect to any clinical variables.

## 4. Discussion

Understanding the pathophysiology of cancer is essential for developing strategies that enable a more personalized approach to MTC. In this context, the investigation of new molecular markers, such as miRNAs, could represent a significant advance in clinical practice, improving both the diagnosis and therapeutic management of this neoplasm.

In our sample, the fold change in miRNA expression was higher in miR-183-5p, followed by miR-375-5p. Similar results were found by Mian et al., where miR-375 was the most overexpressed, followed by miR-183 in third place [22]. Additionally, we observed a higher relative expression of these miRNAs in comparison to paired healthy thyroid tissue, consistent with previous studies [4,14,17,18]. These differences in microRNA expression may be due to malignant transformation and the presence of parafollicular cells (from which this neoplasm derives).

Abraham et al. [16] demonstrated that the expression of miR-21, miR-183, and miR-375 is associated with the worst clinical outcomes, including persistent and metastatic disease.

miR-183-5p is an miRNA that promotes cell proliferation, reduces apoptosis, and enhances migration and invasion of MTC cells. The effect has been linked to the transcription factor FOXO1, which regulates genes involved in apoptosis and DNA damage repair. The overexpression of miR-183-5p is associated with a decrease in FOXO1 levels in MTC [23].

Abraham et al. [16] also found that miR-183-5p overexpression in sporadic MTC was correlated with lymph node involvement, distant metastasis, residual disease, and decreased survival, but no significant differences were observed in familial MTC cases. Similarly, Aubert et al. linked increased miR-183-5p expression to greater tumor aggressiveness, as evidenced by its association with lymph node involvement [15]. This finding was further supported by Gundara et al. [3], who reported a higher expression in tissue from adenopathies affected by MTC, with a positive correlation between the expression of thyroid and nodal tissue.

On the other hand, other authors have not demonstrated that the overexpression of this gene leads to greater aggressiveness [22], as was the case in our study.

Regarding the relationship of miR-375-5p with MTC, different mechanisms have been proposed to explain its influence on MTC prognosis, including the downregulation of YAP1, which promotes tumor progression [4], and dysregulation of the phosphatidylinositol 3-kinase (PI3K)/Akt signaling pathway. Shi et al. [5] identified this pathway as critical for cell growth, apoptosis, cell cycle, metabolism, and angiogenesis; furthermore, its dysregulation adds to the mechanisms promoted by the RET mutation.

Romeo et al. observed that miR-375-5p expression was associated with metastasis, persistence/recurrence, tumor burden, and male sex [18]. Similar results were reported by Censi et al. [10], where patients showed an increase in this miRNA related to males, lymph node involvement, advanced tumor stage, higher Ct levels, and larger tumor size. Galuppini et al. [17] also associated miR-375-5p overexpression with T3/T4, lymph node involvement, stage III/IV, elevated Ct levels at diagnosis, thyroid capsule infiltration, and disease progression. Abraham et al. [16] also showed that miR-375-5p was overexpressed in patients with increased nodal involvement, distant metastases, and residual disease. No correlations were observed with heritability, RET or RAS mutation, age, sex, or Ki67 [4,10,17]. However, other studies, including Mian et al., found no association between miR-375-5p expression and clinical characteristics [22], consistent with our findings.

In reference to miR-144-3p, other studies have shown overexpression in cancer patients [24]. Shabani et al. [19] demonstrated that this miRNA exhibits oncogenic behavior in patients with MTC, showing an association with lymphatic invasion and stages III and IV. However, this was not evident in our study, where the fold change in the expression of this miRNA was lower than that reported by Shabani’s study, which could be attributed to a higher proportion of patients with metastases in their study. Furthermore, Shabani et al. [13] demonstrated increased plasma miR-144-3p levels in hereditary MTC patients. However, in our study, the relative expression of miRNA was lower in tumor tissue than in paired healthy tissue, and no differences in expression were observed according to the clinical characteristics of the patient.

On the other hand, miR-34a-5p has been described as an oncogenic miRNA in both papillary thyroid carcinoma [11] and MTC [19]. However, in another study carried out in MTC, higher expression was not found in patients with worse prognostic factors such as lymph node involvement or the presence of metastases, only in those over 40 years of age [13]. Expression was also not higher in patients with hereditary MTC [19]. Our study also found no significant differences in the fold change in miR-34a-5p expression across clinical characteristics.

miR-200c is a family of miRNAs involved in the epithelial-to-mesenchymal transition and tumor progression in MTC. Specifically, decreased fold change in the expression of miR-200c has been associated with an increase in ZEB-1 and ZEB-2 levels, leading to reduced E-cadherin expression, facilitating invasion and metastasis [23], as demonstrated by Santarpia et al. [25]. On the other hand, Mancikova et al. [26] showed the effect of methylation and observed how hypermethylation resulted in decreased miR-200c expression. In our study, we found a positive correlation with DFS, with a tendency towards lower relative expression in patients with lymph node involvement and persistence/relapse. These findings may be confirmed after increasing the sample size.

The results of our study align with those of previous research, showing a trend towards a lower fold change in the expression in patients with nodal involvement and disease progression. However, no differences in expression were found between sporadic and familial cases.

Regarding the sporadic or familial nature of MTC, some studies have also demonstrated the usefulness of miRNAs, with differential expression of some markers such as miR-9, miR-183, miR-375, miR-144, and miR-34a [13,16]. However, our results show no obvious differences in this regard.

A limitation of this study is the use of healthy thyroid tissue from the same patient rather than from different patients, although some differences were observed. Future studies could consider analyzing healthy thyroid tissue from multiple patients. Furthermore, increasing the sample size would be crucial, as this could help to identify associations with other variables, such as lymph node involvement, as reported in previous studies [26].

miRNA analysis could enhance the evaluation of suspicious nodules in CMT patients. After clinical and ultrasound evaluation of the suspicious nodules, fine-needle aspiration (FNA) cytology usually yields Bethesda category III or category IV. In these cases, miRNA expression analysis may help in the diagnosis of MTC considering an approximate cost of EUR 15. In addition, microRNA analysis could provide guidance on lymph node involvement, aid in decision-making, and offer a more reliable diagnostic approach. Although an in-depth study should be conducted, numerous publications indicate that miRNA screening in cancer is highly cost-effective [27,28,29]. In addition, microRNA analysis could provide guidance on lymph node involvement, decision making, and a more reliable diagnostic approach [30,31,32]. Ciarletto et al. highlighted the utility of a combination miRNA test, including miR-375, to evaluate MTC [31]. The study of miR-183-5p and miR-375 could be useful in the assessment of suspicious adenopathies as they are the most overexpressed miRNAs in our study.

## 5. Conclusions

In conclusion, miRNA 183-5p and miRNA 375-5p are the most highly expressed in MTC patients and may hold potential value for the initial diagnosis of suspicious thyroid nodules.

Higher relative expression of miR-200c-3p was associated with prolonged DFS and lower levels in cases of recurrence or persistence, suggesting that reduced expression of this marker may correlate with greater clinical aggressiveness.

Further research with larger patient cohorts is warranted to confirm and generalize these findings.

## Figures and Tables

**Figure 1 cancers-17-00629-f001:**
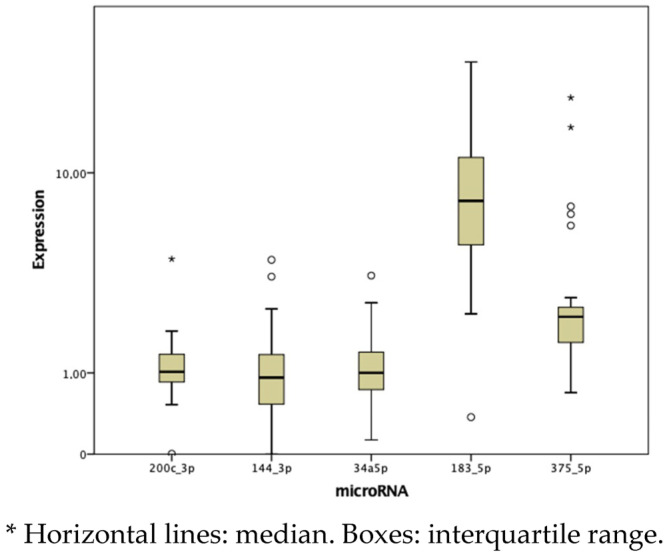
Box plot analysis of the median and interquartile range of miRNAs (2^−ΔΔCt^).

**Figure 2 cancers-17-00629-f002:**
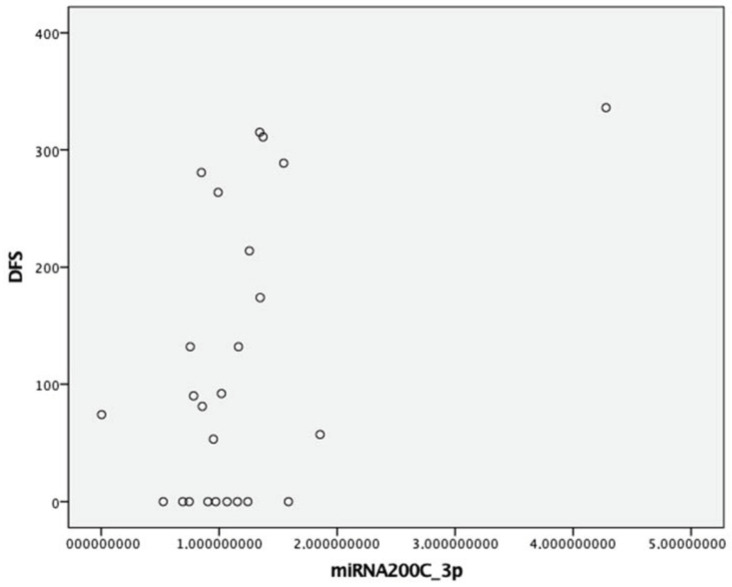
Correlation between DFS and expression of miRNA (2^−ΔΔCt^) 200c-3p.

**Table 1 cancers-17-00629-t001:** Epidemiological and clinical characteristics of patients.

Age at the moment of surgery (*n* = 25)	56,8 years (mean), 43 years (median)
Sex (*n* = 25)	40% (*n* = 10) Male
	60% (*n* = 15) Female
Heredability (*n* = 25)	44% (*n* = 11) Sporadic
	56% (*n* = 14) MEN 2 syndrome
MEN2a syndrome mutations (*n* = 14)	4% (*n* = 1) pC634A
	36% (*n* = 9) pC634T
	12% (*n* = 3) pC634Y
	4% (*n* = 1) L790F
Presurgical calcitonin (*n* = 21)	1556.57 pg/mL (mean), 949 pg/mL (median)
Presurgical CEA (*n* = 21)	30.6 ng/mL (mean), 18.25 ng/mL (median)
Type of surgery (*n* = 25)	28% (*n* = 7) TT
	8% (*n* = 2) TT * and unilateral CND
	4% (*n* = 1) TT * and bilateral CND
	4% (*n* = 1) TT, unilateral CND, and jugular unilateral
	12% (*n* = 3) TT, bilateral CND, and jugular unilateral
	44% (*n* = 11) TT, bilateral CND, and jugular bilateral
Tumour size (mm) (*n* = 24)	23.92 (mean), 20 (median)
Lymphadenectomy (*n* = 25)	72% (*n* = 18) Yes
	28% (*n* = 7) No
Lymph node involvement (*n* = 18)	60% (*n* = 15) Yes
	12% (*n* = 3) No
Number of removed lymph nodes (*n* = 16)	29.19 (mean)
Number of affected lymph nodes (*n* = 15)	7.13 (mean)
Evolution (*n* = 25)	24% (*n* = 6) Cured
	68% (*n* = 17) Persistence
	8% (*n* = 2) Unknown
Structural disease during follow-up (*n* = 25)	52% (*n* = 13) Yes
	48% (*n* = 12) No
DFS (*n* = 25)	115.8 months (mean), 81 months (median)
Exitus (*n* = 25)	20% (*n* = 5) Yes
	80% (*n* = 20) No
Cause of exitus (*n* = 5)	60% (*n* = 3) Neoplastic progression
	40% (*n* = 2) Other causes

* TT: total thyroidectomy; CND: central neck dissection; DFS: disease-free survival.

**Table 2 cancers-17-00629-t002:** TNM classification.

T stage (*n* = 25)	16% (*n* = 4) T1a
	28% (*n* = 7) T1b
	36% (*n* = 9) T2
	16% (*n* = 4) T3
	4% (*n* = 1) Unknown
N stage (*n* = 25)	28% (*n* = 7) Nx
	12% (*n* = 3) N0
	12% (*n* = 3) N1a
	48% (*n* = 12) N1b
M stage (*n* = 25)	100% M0
Stage (*n* = 25)	20% (*n* = 5) Stage I
	12% (*n* = 3) Stage II
	8% (*n* = 2) Stage III
	56% (*n* = 14) Stage IV
	4% (*n* = 1) Unknown

**Table 3 cancers-17-00629-t003:** Relative expression of miRNAs (2^−ΔΔCt^).

*n* = 25	200c_3p	144 3p	34a-5p	183-5p	375-5p
Relative expression (median, IQR *)	1.02 (0.81–1.34)	0.923 (0.45–1.56)	1.002 (0.73–1.43)	7.666 (4.89–13.92)	2.226 (1.46–2.64)

* IQR: interquartile range.

**Table 4 cancers-17-00629-t004:** Comparation of relative miRNA expression in tumor tissue and paired healthy tissue.

Relative Expression (Median)	*n* = 25		
	Tumour Tissue 2^−ΔCt^	Healthy Tissue 2^−ΔCt^	*p*
183-5p	0.061	0.009	0.001
375-5p	0.024	0.009	0.001
34a-5p	0.298	0.26	0.552
200c-3p	6.107	5.11	0.116
144-3p	1.358	1.74	0.046

**Table 5 cancers-17-00629-t005:** Comparison of the relative expression of miRNAs (2^−ΔΔCt^) according to epidemiological and clinical characteristics.

Variables		200c-3p	*p*	144-3p	*p*	34a-5p	*p*	183-5p	*p*	375-5p	*p*
Age at the moment of surgery	Correlation (*n* = 25)	−0.051	0810	0.109	0.603	0.182	0.385	0.179	0.392	−0.138	0.510
Sex	Male (*n* = 10) Female (*n* = 15)	1 1.16	0.579	0.85 1.05	0.346	1.17 0.98	0.912	5.80 7.82	0.244	1.92 2.23	0.202
Heredability	Sporadic (*n* = 11) Men (*n* = 14)	0.90 1.11	0.381	0.74 0.98	0.584	1.33 0.98	0.298	7.82 5.83	0.250	2.16 2.24	0.913
Presurgical calcitonin	Correlation (*n* = 21)	−0.225	0.328	0.135	0.559	−0.187	0.417	−0.377	0.092	0.022	0.924
Lymph node involvement	Yes (*n* = 15) No (*n* = 3) Unknown (*n* = 7)	0.97 1.55 1.26	0.204	0.65 0.89 1.05	0.238	0.98 0.73 1.38	0.124	7.82 6.82 7.67	0.680	1.79 2.23 6.74	0.064
Number of affected lymph nodes	Correlation (*n* = 18)	−0.056	0.824	−0.178	0.479	0.248	0.321	−0.306	0.216	−0.082	0.745
TNM stage	I–II (*n* = 8) III–IV (*n* = 16)	1.36 0.96	0.159	0.98 0.71	0.426	1.29 0.97	0.358	5.96 8.21	0.298	2.36 1.90	0.178
Evolution	Cured (*n* = 6) Persistence (*n* = 17)	1.01 1.02	1	0.86 0.89	0.779	1.17 0.98	0.944	6.83 7.67	0.726	1.46 2.27	1
Structural disease during follow-up	Yes (*n* = 13) No (*n* = 12)	0.97 1.25	0.115	0.92 0.91	0.786	0.96 1.10	0.957	8.83 6.33	0.384	2.25 2.12	0.913
DFS *	Correlation(*n* = 25)	0.400	**0.048**	0.206	0.323	0.172	0.411	−0.204	0.329	0.330	0.108

* DFS: disease-free survival.

## Data Availability

The data presented in this study are available on request from the corresponding author due to (specify the reason for the restriction).

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
