# Peer review of "Could microRNA Analysis Help in the Management of Medullary Thyroid Cancer?"

_cancers, 2025, doi:10.3390/cancers17040629_

Round 1
Reviewer 1 Report
Comments and Suggestions for Authors
Dear Authors, thank you for submitting such an interesting and novel paper to the Cancers journal. Below I present some comments and suggestions to take into consideration before the paper can be processed further:
- Simply summary – I would recommend using a different word rather than ‘on the other hand’. Maybe use the phrase ‘further objectives include’ or ‘the other objective is..’, or something similar
- In the abstract there is no need to use the heading of the particular parts of the abstract such as ‘background/objectives’ or ‘methods’, etc. You can remove these words
- Abstract – in the methods section you mention several micro-RNAs investigated in this study – if all of them are mentioned please use ‘and’ at the end of this enumeration of the microRNAs or if not all of them are mentioned here but just the examples – please use the words ‘etc’ at the end of this enumeration. It will make it clearer for the readers if the number is definite or not.
- ‘Variables studied’ in the abstract – you can use the word ‘included’ instead of ‘:’. Similar to the next sentence; make them the sentences and not the enumeration
- Table 1 seems to be a little bit unclear for me – please consider making it a little bit more reader-friendly as in this form for me it seems to be a little bit chaotic. Maybe it could only be the layout that should be changed to make it clearer
- ‘Regarding tnm classification’ – I suppose it should be indicated as a separate subparagraph
- Table 4 – you can make it a little bit wider
- In the discussion section it would be beneficial to add about the potential costs of the microRNA analysis in terms of using this method as a screening for patients who are at a higher risk of MTC
Best wishes with your further research
Author Response
I would like to thank the editor and reviewers of the journal for the review of our manuscript. We proceed to respond to the change requests below.
Comments 1:Simply summary – I would recommend using a different word rather than ‘on the other hand’. Maybe use the phrase ‘further objectives include’ or ‘the other objective is..’, or something similar
Response 1: We agree with this comment. Therefore, we have modified the phrase accordingly (line 20).
Comments 2: In the abstract there is no need to use the heading of the particular parts of the abstract such as ‘background/objectives’ or ‘methods’, etc. You can remove these words
Response 2: I appreciate your comments, we have removed these headings.
Comments 3: Abstract – in the methods section you mention several micro-RNAs investigated in this study – if all of them are mentioned please use ‘and’ at the end of this enumeration of the microRNAs or if not all of them are mentioned here but just the examples – please use the words ‘etc’ at the end of this enumeration. It will make it clearer for the readers if the number is definite or not.
Response 3: We agree with this comment and have made the necessary adjustment (line 29).
Comments 4: Variables studied’ in the abstract – you can use the word ‘included’ instead of ‘:’. Similar to the next sentence; make them the sentences and not the enumeration
Response 4: Thank you for your suggestion. We have reformulated both sentences to improve clarity (lines 32).
Comments 5: Table 1 seems to be a little bit unclear for me – please consider making it a little bit more reader-friendly as in this form for me it seems to be a little bit chaotic. Maybe it could only be the layout that should be changed to make it clearer
Response 5: We have changed a by the layout as you recommend. Variables have been organized into distinct sections, and the table size has been increased to enhance readability.
Comments 6: Regarding tnm classification’ – I suppose it should be indicated as a separate subparagraph
Response 6: Thank you for your comment, we have added a table with the TNM classification (Table 2) (line 163)
Comments 7: Table 4 – you can make it a little bit wider
Response 7: Of course, we have considered your recommendation.
Comments 8: In the discussion section it would be beneficial to add about the potential costs of the microRNA analysis in terms of using this method as a screening for patients who are at a higher risk of MTC
Response 8: Thank you for your valuable suggestion. We have incorporated this point into the discussion section, addressing the potential costs of microRNA analysis as a screening tool for patients at higher risk of MTC. (line 284-289)

Reviewer 2 Report
Comments and Suggestions for Authors
This paper aims to analyze the expression levels of five microRNAs (miR-183-5p, miR-375-5p, miR-144-3p, miR-34a-5p, miR-200c-3p) in medullary thyroid carcinoma tissue samples and to investigate their association with clinical and pathological features in order to identify potential diagnostic and prognostic biomarkers for MTC.
The subject is relevant but the paper falls short in several aspects and certain questions sholud be addressed.
1) From the Introduction it is not clear why did the authors choose these five microRNAs. These miRs were already analyzed in similar studies/context so what novelty does this particular study bring?
2) The term "genomic study" is not adequate since "genomic" implies high-throughput studies ( such as whole genome sequencing) of a complete set of genetic information.
3) The use of two reference miRs should be described in more detail in the method description. Was geometric mean used or else?
4) In the cited publication (#19), hsa-miR-103a-3p is not listed among the the most sably expressed miRs in any of the tissue analyzed
5) What was the "healthy control"? Corresponding (paired) normal thyroid tissue of each patient or unrelated (unpaired) normal thyroid tissue from one or more patients? This should be explained.
6) Since MTC derives from parafollicular cells, their presence (in more or less differentiated state) in the tumor mass is much greater than in normal thyroid tissue where these cells are scarce while follicular thyroid cells are predominant. So any differences in gene or microRNA expression when comparing MTC tumor tissue with normal thyroid tissue are not just a consequence of malignant transformation but to a great extent a consequence of different cell composition of the tissues analyzed. And these are two very different types of cells (considering embrionic origin, function, cell morphology)... This issue should be addressed in the paper.
Author Response
I would like to thank the editor and reviewers of the journal for the review of our manuscript. We proceed to respond to the change requests below.
Comments 1: From the Introduction it is not clear why did the authors choose these five microRNAs. These miRs were already analyzed in similar studies/context so what novelty does this particular study bring?
Thank you for pointing this out. We selected these microRNAs based on their prior use in similar studies to evaluate whether their findings could be replicated in our cohort of patients. This approach allows us to assess the consistency of these results and confirm their potential diagnostic and prognostic value in MTC
Comments 2: The term "genomic study" is not adequate since "genomic" implies high-throughput studies ( such as whole genome sequencing) of a complete set of genetic information.
We appreciate your observation. We have replaced the term “genomic study” with “miRNA expression analysis” in the manuscript (line 83, 174)
Comments 3: The use of two reference miRs should be described in more detail in the method description. Was geometric mean used or else?
The geometric mean of the reference miRNAs was used for normalization. This information has now been included in the Materials and Methods section (line 91).
Comments 4: In the cited publication (#19), hsa-miR-103a-3p is not listed among the the most sably expressed miRs in any of the tissue analyzed
We sincerely thank the reviewer for their insightful observation. We apologize for the error. The correct reference for stable miRNAs in human cancer tissues is:
Veryaskina, YA, Titov, ET, and Zhimulev, IF. "Reference genes for qPCR-based miRNA expression profiling in 14 human tissues." Med Princ Pract. 2022; 31:322-332 (doi: 10.1159/000524283).
In this review, the authors suggest that miR-103a-3p and miR-191-5p, along with miR-16-5p, can serve as universal reference genes for studying miRNA expression levels across various tissues. We have updated the manuscript accordingly (line 44—references).
Comments 5: What was the "healthy control"? Corresponding (paired) normal thyroid tissue of each patient or unrelated (unpaired) normal thyroid tissue from one or more patients? This should be explained.
The “healthy control” is the corresponding (paired) normal thyroid tissue obtained from the same patients. We clarified this in the revised manuscript (methodology, line 72).
Comments 6: Since MTC derives from parafollicular cells, their presence (in more or less differentiated state) in the tumor mass is much greater than in normal thyroid tissue where these cells are scarce while follicular thyroid cells are predominant. So any differences in gene or microRNA expression when comparing MTC tumor tissue with normal thyroid tissue are not just a consequence of malignant transformation but to a great extent a consequence of different cell composition of the tissues analyzed. And these are two very different types of cells (considering embrionic origin, function, cell morphology)... This issue should be addressed in the paper.
Thank you for this valuable comment. We have addressed the differences in cell composition between MTC tissue and normal thyroid tissue and their potential impact on miRNA expression leves. This discussion has been added to the revised manuscript (lines 210-212).

Reviewer 3 Report
Comments and Suggestions for Authors
The study by Febrero et al. aims to study the expression of selected miRNAs in tissue specimen from medullary thyroid carcinoma (MTC) vs. normal thyroid tissue and to gain valuable information as diagnostic and /or prognostic markers.
Unfortunately this study does add much to the literature, which has addressed this topic since 2008. There are important limitations, including the small, heterogeneous cohort of patients (sporadic and familial forms differ for diagnosis and prognosis), as well as testing of already known, MTC-associated miRNAs. In addition, the adjunct value to diagnosis/prognosis of these miRNAs has not been described by further evaluations, such as ROC curves.
A list of points follow:
TITLE: I would suggest “microRNA”, instead of microrna
ABSTRACT: SLE should be specified.
MATERIALS AND METHODS: The study design should be clearer. The collection period, inclusion and exclusion criteria (if any) should be indicated, as well as the description of the enrolment (consecutive cases?)
Genomic study: I do not agree with this denomination, I would rather define “miRNA expression analysis”. Description of RNA quality controls and purity is lacking. Besides, the chosen miRNAs to normalise results because of their stability are not the ones cited in ref. 19. Anyway, what is important is to choose housekeeping miRNAs that warrant smaller variations across sample sets; this is why normalisation is so critical.
Clinical variables: these could have been described more discursively. Are 6-12 months follow-up enough for prognostic purposes?
RESULTS: Clinical overview: All tables include Spanish wording. Tables are difficult to follow; word breaks should be avoided.
Results from genomic analysis: again, this is not a genomic analysis. Quantitative analysis of miRNA expression in the study cohort, such as Box plot analysis should be also shown. As previously mentioned, besides the association between some miRNAs and MTCs, no attempt has been made to understand whether these miRNAs could add diagnostic/prognostic accuracy to calcitonin (while other papers have explored this important aspect).
DISCUSSION: Discussion should include a more critical point of view, and identify weaknesses and strengths of the study. In addition, limits of the use as normal thyroid tissue as control of MTC should be mentioned and discussed.
REFERENCES: Bibliography should be enriched.
Author Response
I would like to thank the reviewer of the journal for the review of our manuscript. We proceed to respond to the change requests below.
Comments 1: TITLE: I would suggest “microRNA”, instead of microrna
Response 1: We agree with this comment. Therefore, we have revised the tittle and corrected it accordingly (line 2).
Comments 2: ABSTRACT: SLE should be specified.
Response 2: Thank you for your comment, we have corrected this error in the abstract.
Comments 3: MATERIALS AND METHODS: The study design should be clearer. The collection period, inclusion and exclusion criteria (if any) should be indicated, as well as the description of the enrolment (consecutive cases?)
Response 3: We have revised this section to address your concerns. Specifically, we have added the following information (lines 77-80): Twenty-five patients diagnosed with MTC via histological confirmation, operated on by the same surgical team and under follow-up in the Endocrine Surgery Unit of the Hospital Clínico Universitario Virgen de la Arrixaca (HCUVA) during 1998-2016 were included.
Comments 4: Genomic study: I do not agree with this denomination, I would rather define “miRNA expression analysis”. Description of RNA quality controls and purity is lacking. Besides, the chosen miRNAs to normalise results because of their stability are not the ones cited in ref. 19. Anyway, what is important is to choose housekeeping miRNAs that warrant smaller variations across sample sets; this is why normalisation is so critical.
Response 4:
We appreciate your valuable comments, we have revised the text to replace “genomic study” with “miRNA expression analysis” (line 83, 174).
Regarding the normalization references, we apologize for the mistake. The correct reference is: Veryaskina, YA, Titov, ET, and Zhimulev, IF. "Reference genes for qPCR-based miRNA expression profiling in 14 human tissues." Med Princ Pract. 2022; 31:322-332 (doi: 10.1159/000524283). In this review, the authors suggest that miR-103a-3p and miR-191-5p, along with miR-16-5p, can be used as universal reference genes for studying miRNA expression levels across various tissues”.
Comments 5: Clinical variables: these could have been described more discursively. Are 6-12 months follow-up enough for prognostic purposes?
Response 5: We have addressed this issue by revising the methodology to clarify that an initial assessment was conducted at 6-12 months to determine cure or persistence. Subsequently, follow-up continued until 2023 (median follow-up: 204 months; interquartile range: 140-301 months) to evaluate additional variables, including structural disease, metastasis, and disease-free survival (DFS).
Comments 6: RESULTS: Clinical overview: All tables include Spanish wording. Tables are difficult to follow; word breaks should be avoided.
Response 6: Thank you for bringing this to our attention. We have corrected the errors and revised the tables to ensure clarity and consistency.
Comments 7: Results from genomic analysis: again, this is not a genomic analysis. Quantitative analysis of miRNA expression in the study cohort, such as Box plot analysis should be also shown.
Response 7: We agree with this observation. The term “genomic analysis” has been replaced with “miRNA expression analysis” in line 174.
Additionally, we have incorporated a Box plot analysis, which can be found in line 180.
Comments 8: As previously mentioned, besides the association between some miRNAs and MTCs, no attempt has been made to understand whether these miRNAs could add diagnostic/prognostic accuracy to calcitonin (while other papers have explored this important aspect).
Response 8: Thank you for pointing this out. As recommended, we are planning to conduct an analysis comparing the diagnostic performance of miRNAs with pre-surgical calcitonin in a future study. This will require including a larger cohort of patients to achieve greater statistical power.
Comments 9: DISCUSSION: Discussion should include a more critical point of view, and identify weaknesses and strengths of the study. In addition, limits of the use as normal thyroid tissue as control of MTC should be mentioned and discussed.
Response 9: Agree. The discussion section has been revised to address these points. Specifically, the limitation of using normal thyroid tissue as a control has been included on page 7, lines 280-285.
Comments 10: REFERENCES: Bibliography should be enriched.
Response 10: All references concerning microRNA and CMT have been included in the bibliography, as well as some for papillary and follicular thyroid carcinoma. Additionally, we have included information regarding the cost-effectiveness of using microRNA in the diagnostic process (line 59-62, references).

Reviewer 4 Report
Comments and Suggestions for Authors
Dear Dr Ros Madrid
Thank you for your interesting paper. Data regarding the use of new molecular tools towards precision medicine implementation is of great importance to be published.
There are some issues I have highlighted in the manuscript that I believe are very important to be addressed.
Yours sincerely
1. Line 44(...is the differentiated thyroid ...):
This phrase may be confused with DTC. You should rephrase like "MTC is characterized by a more aggressive phenotype compared to DTC especially when distant metastases are present..."
2.Line 45-46(tumours [1])ï¼›
You should add current data https://www.cancer.org/cancer/types/thyroid-cancer/detection-diagnosis-staging/survival-rates.html
3.Line 47(...calcitonin (Ct) levels ...):
Ct levels is a useful presurgical tool. Regarding f-up ct and cea doubling times are the biochemical tools used
4.Line 49(...TNM staging.):
It is not used only in progression. Moreover you should add refs. here
5.Line 101(preoperative Ct levels, sporadic or familial):
CEA levels should be added as well
6. Line 107(their Ct levels at 6–12 months):
CEA levels should be added as well
7.Table 1. Epidemiological and clinical characteristics of patients.
Please use English language in the whole table
8. Line 150(...and 8% (n=2) had unknown outcomes):
Do you mean lost to f-up?
9. Line 151(At follow-up, 52% (n=13)...):
As soon as you have the data of 23 patients you should change this percentage accordingly
10.Line 152-153(un-known):
lost to f-up?
11.Line 153-154(During follow-up, 20% (n=5) of patients 153 died:):
Again you should modify the statistics in regard to the patients you do have data for f-up.
12.Line 156(3.2 Results of genomic analysis):
Statistics should be modified in regard to the patients you have data for f-up. Otherwise it is not safe to talk about correlations between specific miRNAs and outcome as soon as you don't have the data for the outcome.
Author Response
I would like to thank the reviewer of the journal for the review of our manuscript. We proceed to respond to the change requests below.
Comments 1: Line 44(...is the differentiated thyroid ...):
This phrase may be confused with DTC. You should rephrase like "MTC is characterized by a more aggressive phenotype compared to DTC especially when distant metastases are present..."
Response 1: We agree with this comment. Therefore, we proceed to modify it (line 48)
Comments 2: Line 45-46 (tumours [1]). You should add current data https://www.cancer.org/cancer/types/thyroid-cancer/detection-diagnosis-staging/survival-rates.html
Response 2: Thank you for the suggestion. We have updated the manuscript by incorporating the most recent survival rate data from the suggested source (lines 46–47).
Comments 3:.Line 47(...calcitonin (Ct) levels ...): Ct levels is a useful presurgical tool. Regarding f-up ct and cea doubling times are the biochemical tools used
Response 3: We appreciate this comment and have included information about Ct and CEA doubling times as important biochemical tools for follow up (line 50-53).
Comments 4: Line 49(...TNM staging.):It is not used only in progression. Moreover you should add refs. Here
Response 4: We agree with this observation. We have clarified the use of TNM staging and added appropriate references to support the statement (line 52-53).
Comments 5: Line 101(preoperative Ct levels, sporadic or familial): CEA levels should be added as well
Response 5: Of course, we have added CEA levels in both the text and Table 1 and line 135-136.
Comments 6: Line 107(their Ct levels at 6–12 months): CEA levels should be added as well. In the same way, we have added this information in the Table
Response 6: As suggested, we have included CEA levels in the revised text (line 169-170).
Comments 7: Table 1. Epidemiological and clinical characteristics of patients.
Please use English language in the whole table
Response 7: Thank you for your comment, we have corrected this and ensured that all table content is in English.
Comments 8: Line 150(...and 8% (n=2) had unknown outcomes): Do you mean lost to f-up?
Response 8: Thank you for your observation. These two patients were not evaluated for calcitonin levels at 6–12 months, so we cannot classify them as having persistent disease or being cured. However, we believe their inclusion in the study is important because we have data on their long-term clinical evolution. Both patients had structural disease during follow-up, but we cannot differentiate whether it represents persistent disease or recurrence. These cases are included in both the structural recurrence and DFS variables. We have clarified this point in the Methods section (lines 114–116).
Comments 9: Line 151(At follow-up, 52% (n=13)):
As soon as you have the data of 23 patients you should change this percentage accordingly
Response 9: As justified in the response to comment 8, these 2 patients are not excluded from the analysis, and the percentages remain inchanged.
Comments 10: Line 152-153(un-known):lost to f-up?
Response 10: As explained previously, these patients were not lost to follow-up but were not evaluated for calcitonin levels at 6–12 months. We have clarified this in the revised manuscript.
Comments 11: Line 153-154(During follow-up, 20% (n=5) of patients 153 died:): Again you should modify the statistics in regard to the patients you do have data for f-up.
Response 11: As addressed in the response to Comment 8, the two patients with unknown 6–12-month calcitonin results are not excluded, and the data and percentages remain unchanged.
Comments 12: Line 156(3.2 Results of genomic analysis):
Statistics should be modified in regard to the patients you have data for f-up. Otherwise it is not safe to talk about correlations between specific miRNAs and outcome as soon as you don't have the data for the outcome.
Response 12: As justified in question 8, these 2 patients are not discarded and the same data are kept.

Round 2
Reviewer 1 Report
Comments and Suggestions for Authors
Dear Authors,
thank you very much for correcting the manuscript according to my comments and suggestions.
I have no further comments regarding this paper.
I wish you all the best with your further research
With kind regards
A reviewer
Author Response
Revisor 1
Dear Authors,
thank you very much for correcting the manuscript according to my comments and suggestions.
I have no further comments regarding this paper.
I wish you all the best with your further research
With kind regards
A reviewer
Response: Thank you very much to the revisor for his comments and appreciations.
Reviewer 2 Report
Comments and Suggestions for Authors
I have some additional suggestions for improving this manuscript:
1) The therm "corresponding" or "paired" " normal thyroid tissue" sholud be used throughout the whole text of the manuscript
2) In the Abstract (line 34) the authors report "mean expression" of microRNAs but there is no data of it in the manuascript.
3) In the Methods, some minimum information on how microRNA analysis was conducted- what reagents were used, report primers (assays) for each microRNA, RT product quantity, reaction conditions for real-time PCR, number of reaction (technical)replicates...
4) When presenting results, measures of variability of data should be presented, as mean±SD or median with quartiles.
5) Figure 1 and Supplementary material are nor refered to anywhere in the text. Figure 1 sholud be described- what do boxes, horizontal lines and error bars represent
6) Line 174- In relative quantification- the results 2 – ΔΔCt represent expression fold change relative to the corresponding normal thyroid tissue (after normalization to endogenous control), not the expression itself but the alterations, so the term " highest expression" should be rephrazed accordingly. See ref#21 in the manuscript and
Schmittgen, T., Livak, K. Analyzing real-time PCR data by the comparative CT method. Nat Protoc 3, 1101–1108 (2008). https://doi.org/10.1038/nprot.2008.73
Same goes for Fig1 graph, Conclusion (line 299) and wherever mentioned in this manner
7) In Table 4 results are presented only for 14 sample pairs. This should be pointed out and explained which samples are those
8) The authors might consider presenting some of the results as scatter dot plots
Author Response
I have some additional suggestions for improving this manuscript:
1) The therm "corresponding" or "paired" " normal thyroid tissue" sholud be used throughout the whole text of the manuscript
Response 1: Thank you for your valuable suggestion. We have incorporated paired normal thyroid tissue in the whole text. (line 36, 99, 105, 109, 184, 224, 271)
2) In the Abstract (line 34) the authors report "mean expression" of microRNAs but there is no data of it in the manuascript.
Response 2: We sincerely thank the reviewer for their insightful observation. We apologize for the error. We have changed the mean expression for median (line 34-35)
3) In the Methods, some minimum information on how microRNA analysis was conducted- what reagents were used, report primers (assays) for each microRNA, RT product quantity, reaction conditions for real-time PCR, number of reaction (technical)replicates...
Response 3: Thank you for pointing this out. We have completed the methods with all the details that you recommend (line 86-88 and 91-96)
4) When presenting results, measures of variability of data should be presented, as mean±SD or median with quartiles.
Response 4: We agree with this observation. We have included de interquartile range in table 3.
5) Figure 1 and Supplementary material are nor refered to anywhere in the text. Figure 1 sholud be described- what do boxes, horizontal lines and error bars represent
Response 5: We agree with this observation. We have described as you recommend (line 179, 183-185)
6) Line 174- In relative quantification- the results 2 – ΔΔCt represent expression fold change relative to the corresponding normal thyroid tissue (after normalization to endogenous control), not the expression itself but the alterations, so the term " highest expression" should be rephrazed accordingly. See ref#21 in the manuscript and
Schmittgen, T., Livak, K. Analyzing real-time PCR data by the comparative CT method. Nat Protoc 3, 1101–1108 (2008). https://doi.org/10.1038/nprot.2008.73
Same goes for Fig1 graph, Conclusion (line 299) and wherever mentioned in this manner
Response 6: We appreciate your valuable comments, we have changed the whole text as you recommended. You can find it in the line 177,178,180, 195, 198, 218, 263, 272, 277, 279, 288, .
7) In Table 4 results are presented only for 14 sample pairs. This should be pointed out and explained which samples are those
Response 7: We sincerely thank the reviewer for the observation. We apologize for the error. We have correct it (table 4)
8) The authors might consider presenting some of the results as scatter dot plots
Response 8: Thank you for the recommendation. We have included the figure 2 with scatter dot plots of the disease free survival and relative expression of miRNA 200c-3p (line 208-209)
Reviewer 3 Report
Comments and Suggestions for Authors
the authors have sufficiently addressed the concerns raised.
Round 3
Reviewer 2 Report
Comments and Suggestions for Authors
With all due respect for the authors and their effort, I must conclude that they failed to address some critical issues in their manuscript.
These are quite basic things such as graphical representation of the results (the explanation of the error bars in Figure 1 is still missing, Figure 2 actually shows that there is no correlation between DFS and miR-200c-3p indicaing misinterpretation of the statistical analysis results)...
There are also issues with basic terminology ("fold change expression" is not the same as "expression fold change"), indicating that there is some lack of understanding of the results that are obtained by comparative Ct method, thus affecting the interpretation of the results, as well as the conclusions drawn based on them.
Although many issues have been addressed, some major flaws are still there and the overall quality of the manuscript has not improved enough after two revisions. Therefor I recommend rejection.